ecology, computational biology, structural biology

COVID-19, ecological traits, zoonotic, spillback, machine learning, structural modelling

**Author for correspondence:**
Barbara A. Han
e-mail: hanb@caryinstitute.org

†Contributed equally.

# Predicting the zoonotic capacity of mammals to transmit SARS-CoV-2

Ilya R. Fischhoff[1,†], Adrian A. Castellanos[1,†], João P. G. L. M. Rodrigues[2], Arvind Varsani[3,4] and Barbara A. Han[1]

[1]Cary Institute of Ecosystem Studies, Box AB Millbrook, NY 12545, USA
[2]Department of Structural Biology, Stanford University School of Medicine, Stanford, CA 94305, USA
[3]The Biodesign Center for Fundamental and Applied Microbiomics, Center for Evolution and Medicine, School of Life Sciences, Arizona State University, Tempe, AZ 85287, USA
[4]Structural Biology Research Unit, Department of Integrative Biomedical Sciences, University of Cape Town, 7700 Cape Town, Rondebosch, South Africa

IRF, 0000-0001-6956-8284; AAC, 0000-0002-3412-0487; JPGLMR, 0000-0001-9796-3193; AV, 0000-0003-4111-2415; BAH, 0000-0002-9948-3078

Back and forth transmission of severe acute respiratory syndrome coronavirus 2 (SARS-CoV-2) between humans and animals will establish wild reservoirs of virus that endanger long-term efforts to control COVID-19 in people and to protect vulnerable animal populations. Better targeting surveillance and laboratory experiments to validate zoonotic potential requires predicting high-risk host species. A major bottleneck to this effort is the few species with available sequences for angiotensin-converting enzyme 2 receptor, a key receptor required for viral cell entry. We overcome this bottleneck by combining species' ecological and biological traits with three-dimensional modelling of host-virus protein–protein interactions using machine learning. This approach enables predictions about the zoonotic capacity of SARS-CoV-2 for greater than 5000 mammals—an order of magnitude more species than previously possible. Our predictions are strongly corroborated by *in vivo* studies. The predicted zoonotic capacity and proximity to humans suggest enhanced transmission risk from several common mammals, and priority areas of geographic overlap between these species and global COVID-19 hotspots. With molecular data available for only a small fraction of potential animal hosts, linking data across biological scales offers a conceptual advance that may expand our predictive modelling capacity for zoonotic viruses with similarly unknown host ranges.

## 1. Introduction

The ongoing COVID-19 pandemic has surpassed 4.8 million deaths globally as of 1 October 2021 [1,2]. Like previous pandemics in recorded history, COVID-19 originated from the spillover of a zoonotic pathogen, severe acute respiratory syndrome coronavirus 2 (SARS-CoV-2), a betacoronavirus originating from an unknown animal host [3–6]. The broad host range of SARS-CoV-2 is due in part to its use of a highly conserved cell surface receptor to enter host cells, the angiotensin-converting enzyme 2 receptor (ACE2) [7] found in all major vertebrate groups [8].

The ubiquity of ACE2 coupled with the high prevalence of SARS-CoV-2 in the global human population explains multiple observed *spillback* infections since the emergence of SARS-CoV-2 in 2019 (see natural infections listed in electronic supplementary material, table S1). In spillback infection, human hosts transmit SARS-CoV-2 virus to cause infection in non-human animals. In addition to threatening wildlife and domestic animals, repeated spillback infections may lead to the establishment of new animal hosts from which SARS-CoV-2 can then pose a risk of *secondary spillover* infection to humans

through bridge hosts (e.g. [9]) or newly established enzootic reservoirs. Indeed, this risk has been realized in Denmark [10] and The Netherlands, where SARS-CoV-2 spilled back from humans to farmed mink (*Neovison vison*) with the secondary spillover of a SARS-CoV-2 variant from mink back to humans [11]. A major concern in such secondary spillover events is the appearance of a mutant strain affecting host range [12] or leading to increased transmissibility in humans [13,14] (but see [15,16]), reduced sensitivity to neutralizing antibodies [10] and reduced vaccine efficacy [17]. Conversely, human-derived variants pose spillback risks to animals. For example, in contrast with previous infection trials [18], two new human variants have overcome the species barrier to infect laboratory mice (*Mus musculus*) [19].

Spillback infections from humans to animals are already occurring worldwide with pets, domesticated animals, zoo animals and wildlife now documented as new hosts of SARS-CoV-2 (figure 1; electronic supplementary material, table S1). SARS-CoV-2 has been found for the first time in wild and escaped mink in multiple states in the United States, with viral sequences identical to SARS-CoV-2 in nearby farmed mink [33–35]. The global scale of human infections and the increasing range of known animal hosts demonstrate that SARS-CoV-2 has the capacity to establish novel infection cycles in animals. In response, recent computational studies predict the susceptibility of particular animal species to SARS-CoV-2 [12,21–28,36] by comparing known sequences of ACE2 orthologues across species (*sequence-based* studies), or by modelling the structure of the viral spike protein bound to ACE2 orthologues (*structure-based* studies) to yield a wide range of predictions with varying degrees of agreement with laboratory animal experiments (figure 1).

Sequence-based studies predict host susceptibility based on amino acid sequence similarity between human (hACE2) and non-human ACE2, and assume that a high degree of similarity correlates with stronger viral binding, especially at amino acid residues where hACE2 interacts with the SARS-CoV-2 spike glycoprotein. For some species, such as rhesus macaques [37], these qualitative predictions are borne out by *in vivo* studies (figure 1), but predictions from these methods do not consistently match real-world outcomes. For example, sequence similarity predicted weak viral binding for minks and ferrets, which have both been confirmed as highly susceptible [11,22,38] (figure 1). Mismatches to *in vivo* outcomes may arise in part because protein three-dimensional structure, the main determinant of the interaction between host ACE2 and the viral spike protein, is incompletely represented by one-dimensional amino acid sequences [39,40].

Structure-based studies model the three-dimensional structure of protein-protein complexes to address some of the limitations of sequence-based approaches. Structural models have proven useful for predicting how different ACE2 orthologues bind to the SARS-CoV-2 viral spike protein receptor-binding domain (RBD) [12,28]. These studies leverage known structures of the hACE2 receptor bound to the SARS-CoV-2 RBD and use powerful simulations to predict how variation across different ACE2 orthologues affects binding with the viral RBD. While these approaches successfully predicted strong binding for species that have been infected (e.g. domestic cat, tiger, dog and ferret) and weak binding for species in which experimental infections have failed (e.g. chicken, duck [38], mouse [18]), the results are also not consistently supported by experiments. For instance,

while guinea pig ACE2 scored favourably in one structure-based study [12], this orthologue was shown experimentally not to bind to the SARS-CoV-2 RBD [29].

Although structural modelling has produced the most accurate results to date, all currently available approaches for predicting the host range of SARS-CoV-2 are fundamentally constrained by the availability of ACE2 sequences across species. ACE2 is ubiquitous across chordates, probably because of its role in highly conserved physiological pathways, for example in regulating blood pressure, salt and water [41]. The vast majority of mammal species (greater than 6000 species) are likely to have ACE2, but sequences are available for only around 300 species. The functional importance of the ACE2 receptor suggests that it has evolved in association with other intrinsic organismal traits for which data are available for many more species. These suites of correlated organismal traits may provide a robust statistical proxy that can be leveraged to predict biologically permissive hosts for SARS-CoV-2. Previous trait-based analyses applied machine learning techniques to accurately distinguish the zoonotic capacity of various organisms [42–44] and predict likely hosts for particular groups of related viruses [45,46], predictions which have subsequently been validated through independent laboratory and field investigations (e.g. [47,48]).

Here, we combine structural modelling of viral binding with machine learning of species ecological and biological traits to predict zoonotic capacity for SARS-CoV-2 across 5400 mammal species, expanding our predictive capacity by an order of magnitude (figure 2). Crucially, this integrated approach enables predictions for the vast majority of species whose ACE2 sequences are currently unavailable by leveraging information from viral binding dynamics and biological traits. In our workflow (figure 2), we first carry out structural modelling to quantify the binding strength of SARS-CoV-2 RBD for vertebrate species using published ACE2 amino acid sequences [49]. We then collate species traits and train a machine learning model to predict the zoonotic capacity for 5400 mammals.

As COVID-19 is primarily a disease affecting humans, spillback infection of SARS-CoV-2 from humans to animals is the most likely mode by which new animal hosts will become established. We therefore identify a subset of species for which the threat of spillback infection appears greatest due to geographic overlaps and opportunities for contact with humans in areas of high SARS-CoV-2 prevalence globally. These approaches underscore the utility of establishing interdisciplinary and iterative processes that join computational modelling, field surveillance and laboratory experiments to more efficiently quantify zoonotic risk [50], and better inform next steps to prevent enzootic SARS-CoV-2 transmission and spread. Our analyses are based on the initial dominant SARS-CoV-2 variant in humans, but these methods can be readily adjusted to enable host range predictions for new variants as their hACE2-RBD crystal structures become available.

## 2. Methods

### (a) Structural modelling of ACE2 orthologues bound to SARS-CoV-2 spike

We assembled ACE2 sequences from the NCBI GenBank and MEROPS databases. The modelling of 326 ACE2 orthologues bound to SARS-CoV-2 spike RBD was carried out using the

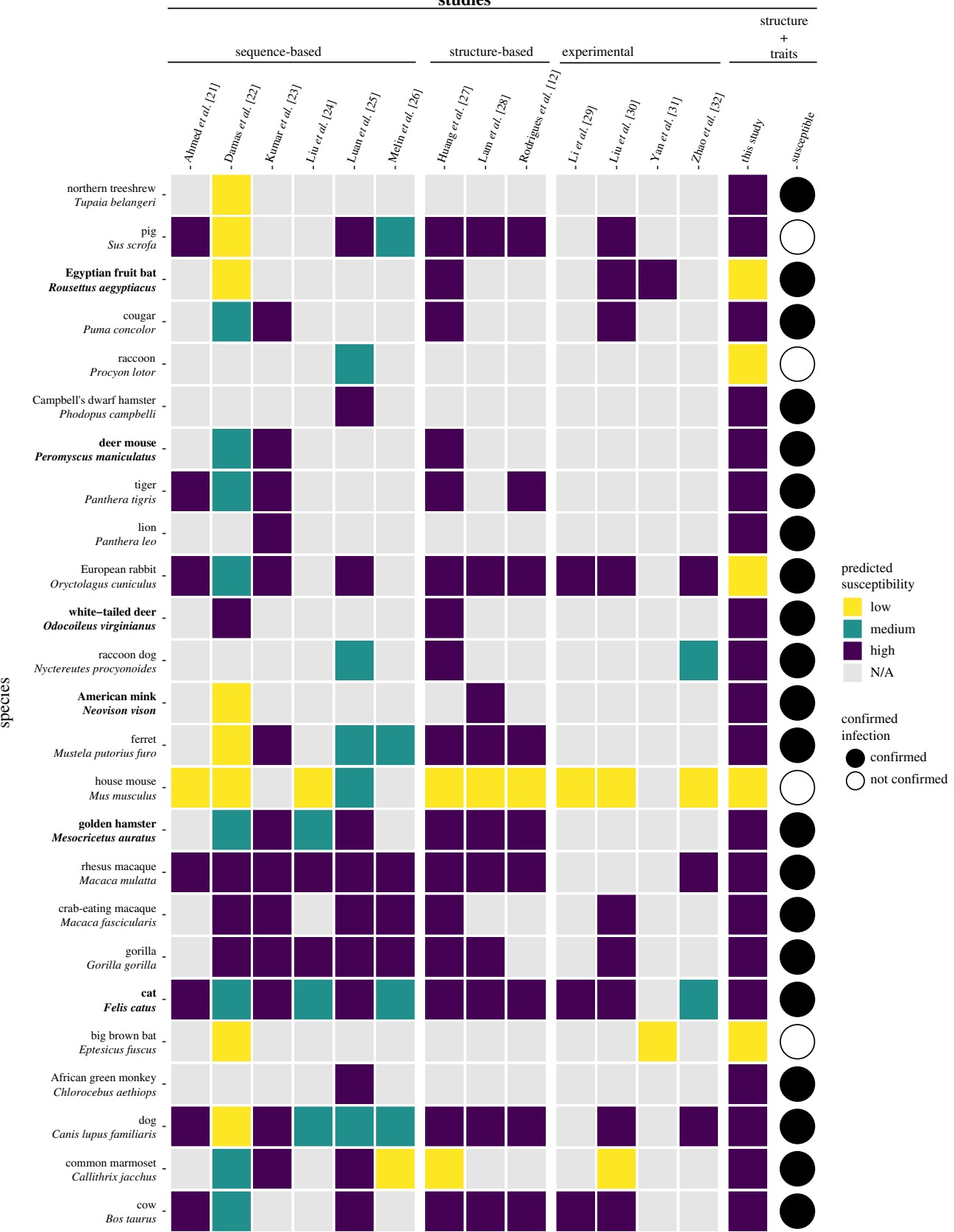

**Figure 1.** A heatmap summarizing predicted susceptibility to SARS-CoV-2 for species with confirmed infection from *in vivo* experimental studies or from documented natural infections. Studies that make predictions about species susceptibility are shown on the *x*-axis, organized by the method of prediction (those relying on ACE2 sequences, estimating binding strength using three-dimensional structures, or laboratory experiments). Predictions about zoonotic capacity from this study are listed in the second to last column, with high and low categories determined by zoonotic capacity observed in *Felis catus*. Confirmed infections for species along with the *y*-axis are depicted as a series of filled or unfilled circles. Bolded species have been experimentally confirmed to transmit SARS-CoV-2 to naive conspecifics. Species predictions range from warmer colours (yellow: low susceptibility or zoonotic capacity for SARS-CoV-2) to cooler colours (purple: high susceptibility or zoonotic capacity). See electronic supplementary material, Methods [20] for detailed methods about how predictions from past studies were categorized as low, medium or high. For a comparison of predictions of species susceptibility from multiple methods, including our study, see electronic supplementary material, figure S1. (Online version in colour.)

**Figure 2.** A flowchart showing the progression of our workflow combining evidence from limited laboratory and field studies with additional data types to predict zoonotic capacity across mammals through multi-scale statistical modelling (grey boxes, steps 1–5). For all vertebrates with published ACE2 sequences, we modelled the interface of species' ACE2 bound to the viral RBD using HADDOCK. We then combined the HADDOCK scores, which approximate binding strength, with species' trait data and trained machine learning models (generalized boosted regression) for both mammals and vertebrates. Predictive modelling of host zoonotic capacity focused on mammals only because there are currently no non-mammalian hosts for SARS-CoV-2 and imbalanced ACE2 sequences among non-mammals. Mammal species predicted to have high zoonotic capacity were then compared to results of *in vivo* experiments and in silico studies that applied various computational approaches. Based on predictions from our model, we identified a subset of species with particularly high risk of spillback and secondary spillover potential to prioritize additional laboratory validation and field surveillance (dashed line). (Online version in colour.)

HADDOCK software as described previously [13], with a few differences. For details on how we processed sequences, and on our structural modelling of ACE2 orthologues bound to SARS-CoV-2 spike (PDB ID: 6m0j [51]; see electronic supplementary material, Methods [20]). For each species, we estimated binding strength based on HADDOCK score—a combination of van der Waals, electrostatics and desolvation energies. A lower (more negative) HADDOCK score predicts stronger binding between the two proteins. We hereafter refer to predicted binding strength, or simply binding strength, to indicate HADDOCK score. The HADDOCK server is freely available, and we provide code to reproduce analyses or to aid in the application of this modelling approach to similar problems [52]. Though the effects of multiple simultaneous mutations on binding affinity remain difficult to predict, HADDOCK has been used to explore how minor changes to the structure of the RBD (e.g. from point mutations or deletions in SARS-CoV-2 variants) affect viral binding [12]. These variations are accounted for by the HADDOCK software by estimating and considering their effects on both strong and weak forces that together determine both local and interfacial molecular contacts [53]. As crystal structures of novel variants become available, the modelling pipeline we present here can be applied to predict how differences in binding strengths to ACE2 orthologues affect host range and the risk of spillover transmission across species.

## (b) Trait data collection and cleaning
We gathered ecological, life history, phylogenetic and biological trait data from AnAge [54], Amniote Life History Database [55], EltonTraits [56], PanTHERIA [57] and taxonomic databases [58], among other databases. Non-mammal hosts have yet to be confirmed as both susceptible and capable of onward transmission of SARS-CoV-2. Therefore, while we gathered data on certain traits across all vertebrates, we gathered data on additional traits for mammals. For mammal species, we applied boosted regression

(BRT) using the gbm package [59] in R v. 4.0.0 [60] to impute missing trait data (e.g. [44]; see electronic supplementary material, Methods for imputation methods and results). Data and descriptions of each variable can be found in the electronic supplementary material, table S2. For details on data processing, see electronic supplementary material, Methods [20].

## (c) Modelling
### (i) Quantifying a threshold for zoonotic capacity using HADDOCK
ACE2 binding is necessary for viral entry into host cells, but it is not sufficient for SARS-CoV-2 transmission. Multiple *in vivo* experiments suggest that some species are capable of binding SARS-CoV-2 but not capable of transmitting active infection to other individuals (e.g. cattle, *Bos taurus* [61]; bank voles, *Myodes glareolus* [62]). Viral replication and viral shedding that enables onward transmission are both required to become a suitable bridge or reservoir host for SARS-CoV-2. We constrained our predictions to species with the greatest potential for onward transmission by training our models on a binary label created using a conservative threshold of binding strength (HADDOCK score = −129). This value falls between the scores for two species: the domestic cat (*Felis catus*), currently the species with weakest predicted binding but confirmed conspecific transmission [63], and the pig (*Sus scrofa*), which shows the strongest estimated binding for which experimental inoculation failed to cause detectable infection [38]. Binding strength was binarized according to this threshold, above which it is more likely that both infection and onward transmission will occur following the results of multiple empirical studies (figure 1; electronic supplementary material, table S1). There are susceptible species whose predicted binding strength is weaker than cats, but conspecific transmission has not been confirmed in these species. For additional modelling details, see electronic supplementary material, Methods.

## (d) Trait-based modelling to predict zoonotic capacity

We applied generalized BRT [64] to host trait data to predict mammal species' zoonotic capacity (for descriptions and results from additional uninformative models, including models using a vertebrate dataset, see electronic supplementary material, Methods and electronic supplementary material table S3). Code for BRT modelling, along with data for training models and making predictions, can be found at https://github.com/Han-LabDiseaseEcology/zoonotic_capacity.

## 3. Results

Currently available ACE2 protein sequences came from 326 species spanning eight classes and 87 orders [52]. The majority of sequences belonged to the classes Actinopterygii (22.1%), Aves (23.3%) and Mammalia (46.6%). We predicted binding strength for 299 vertebrates, including 142 mammals (electronic supplementary material, figures S2–S6). Among well-represented mammalian orders (those containing at least 10 species with binding strength predictions), Primates and Carnivora showed predicted mean binding strengths that were stronger than domestic cats (electronic supplementary material, figure S2).

We next constructed a trait-based machine learning model to predict zoonotic capacity (a binarized binding threshold) in mammals. We used the best-performing model to generate predictions of zoonotic capacity among mammal species (corrected test AUC of 0.72; for results of all other model variations see electronic supplementary material, table S3). Citation count in Web of Science, used as a proxy for study effort, had approximately 1% relative importance (electronic supplementary material, figure S7), suggesting that sampling bias across species had little influence on the model.

This model predicted 540 species spanning 13 orders to have zoonotic capacity within the 90th percentile (0.826 or higher, with a total of 2401 mammal species with prediction scores above 0.5; see electronic supplementary material, file S1 for predictions on all 5400 species [20]). Most primates were predicted to have high zoonotic capacity and collectively showed stronger viral binding compared to other mammal groups (figure 3). Additional orders predicted to have high zoonotic capacity (at least 75% of species above 0.5) include Hyracoidea (hyraxes), Perissodactyla (odd-toed ungulates), Scandentia (treeshrews), Pilosa (sloths and anteaters), Pholidota (pangolins) and non-cetacean Artiodactyla (even-toed ungulates) (figure 3). Results of model bootstrap iteration predictions can be found in the electronic supplementary material, file S1 and electronic supplementary material figure S10.

## (a) Comparing model predictions to *in vivo* outcomes

These model predictions matched the experimental infection outcomes of several *in vivo* studies (figure 1). For instance, experiments on deer mice (*Peromyscus maniculatus* [65,66]) and raccoon dogs (*Nyctereutes procyonoides* [67]) confirmed SARS-CoV-2 infection and transmission to naive conspecifics. Our model also estimated a high probability of zoonotic capacity of American mink for SARS-CoV-2 (*Neovison vison*, probability = 0.83, 90th percentile), in which farmed individuals present severe infection and demonstrate the capacity to transmit to conspecifics as well as to humans [11]. Our

model also correctly predicted relatively low zoonotic capacity for big brown bats (*Eptesicus fuscus* [68]).

There were notable differences between our model results and currently available experimental studies. Our model estimated high zoonotic capacity for pigs (*Sus scrofa*, probability = 0.72, approximately 80th percentile), but *in vivo* studies report no detectable infection or onward transmission [38,69]. Similarly for cattle (*Bos taurus*), our model estimated a moderately high probability for zoonotic capacity (0.72, approximately 80th percentile), whereas in a live animal experiment, cattle were susceptible but with no onward transmission to conspecifics [61].

## 4. Discussion

We combined structure-based models of viral binding with species-level data on biological and ecological traits to predict the capacity of mammal species to become zoonotic hosts of SARS-CoV-2 (*zoonotic capacity*). Importantly, this approach extends our predictive capacity beyond the limited number of species for which ACE2 sequences are currently available. Numerous mammal species were predicted to have zoonotic capacity that meets or exceeds the viral susceptibility and transmissibility observed in experimental infections with SARS-CoV-2 (figure 1; electronic supplementary material, table S1). Many species with high model-predicted zoonotic capacity also live in human-associated habitats and overlap geographically with global COVID-19 hotspots (figure 4). Below we discuss predictions of zoonotic capacity for a number of ecologically and epidemiologically relevant categories of mammalian hosts.

## (a) Captive, farmed or domesticated species

Given that contact with humans fundamentally underlies transmission risk, it is notable that our model predicted high zoonotic capacity for multiple captive species that have also been confirmed as susceptible to SARS-CoV-2. These include numerous carnivores, such as large cats from multiple zoos and pet dogs and cats. Our model also predicted high SARS-CoV-2 zoonotic capacity for many farmed and domesticated species. The water buffalo (*Bubalus bubalis*), widely kept for dairy and plowing, had the highest probability of zoonotic capacity among livestock (0.91). Model predictions in the 90th percentile also included American mink (*Neovison vison*), red fox (*Vulpes vulpes*), sika deer (*Cervus nippon*), white-lipped peccary (*Tayassu pecari*), nilgai (*Boselaphus tragocamelus*) and raccoon dogs (*Nyctereutes procyonoides*), all of which are farmed. The escape of farmed individuals into wild populations has implications for the enzootic establishment of SARS-CoV-2 [33]. These findings also have implications for vaccination strategies, for instance, prioritizing people in contact with potential bridge species (e.g. slaughterhouse workers, farmers, veterinarians).

## (b) Live traded or hunted wildlife species

The *Macaca* genus comprised the majority of live-traded primates. Our model predicted high zoonotic capacity for all *Macaca* species (20/21 species, with all species within the top 10% of predictions except *M. assamensis*). Several live-traded carnivores and pangolins were also assigned high zoonotic capacity, including the Asiatic black bear (*Ursus*

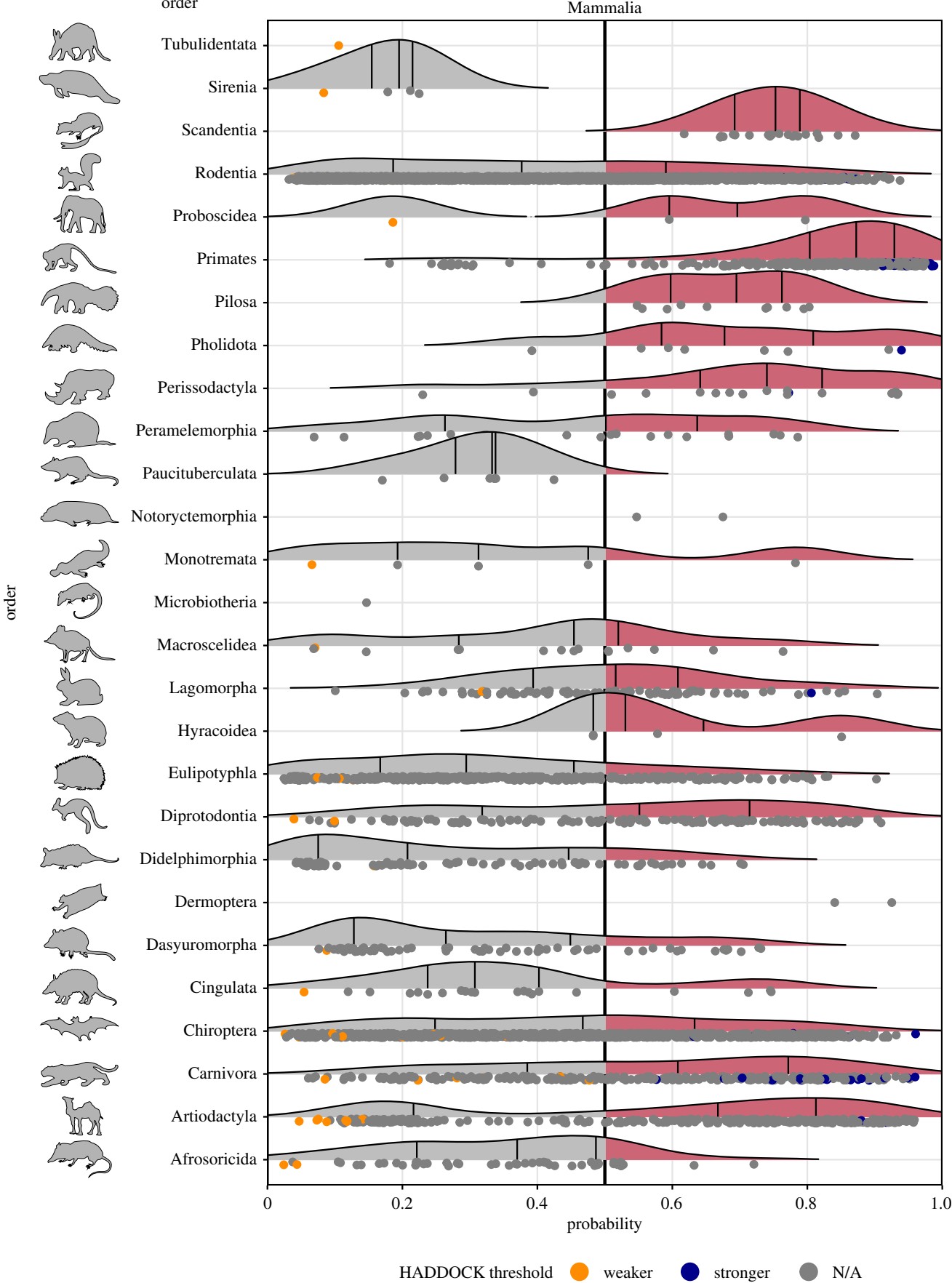

**Figure 3.** Ridgeline plots showing the distribution of predicted zoonotic capacity across mammals. Predicted probabilities for zoonotic capacity across the *x*-axis range from 0 (likely not susceptible) to 1 (zoonotic capacity predicted to be the same or greater than *Felis catus*), with the vertical line representing 0.5. The *y*-axis depicts all mammalian orders represented by our predictions. Density curves represent the distribution of the predictions, with those parts of the curve over 0.5 coloured pink and lines representing distribution quartiles. The predicted values for each order are shown as points below the density curves. Points that were used to train the model are coloured: orange represents species with weaker predicted binding, blue represents species with stronger predicted binding. Selected family-level distributions are shown in the electronic supplementary material, figures S8 and S9 [20]. (Online version in colour.)

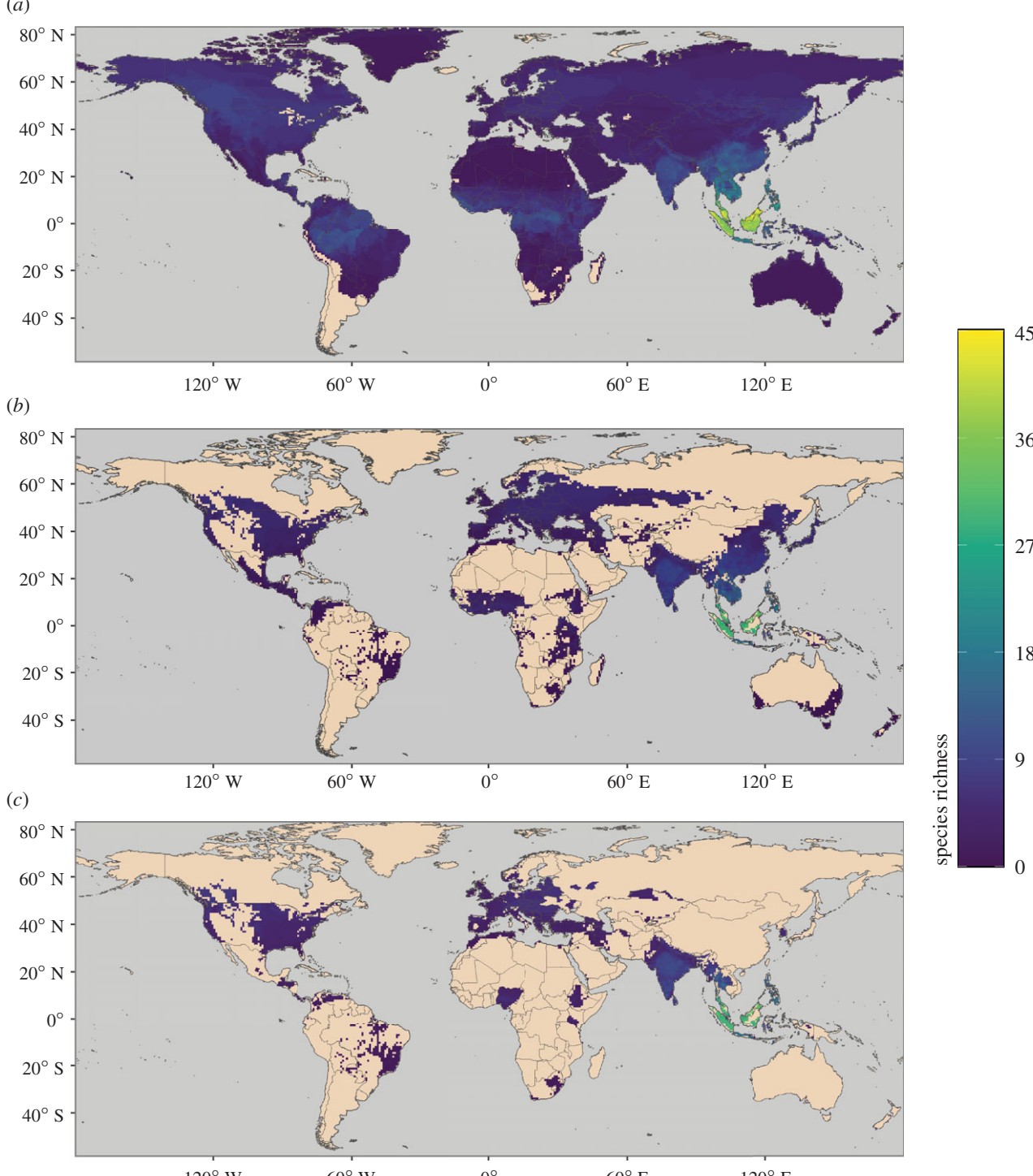

**Figure 4.** Maps showing the global distribution of species with the predicted capacity to transmit SARS-CoV-2; (*a*) depicts global species richness of the top 10 per cent of model-predicted zoonotic capacity. Geographic ranges of this subset of species were filtered to those associated with human-dominated or human-altered habitats (*b*), and further filtered to show the subset of species that overlaps with areas of high human SARS-CoV-2 positive case counts (over 100 000 cumulative cases as of 17 May 2021) (*c*). For a full list of model-predicted zoonotic capacity of species by country, see electronic supplementary material, file S2 [20]. (Online version in colour.)

*thibetanus*), grey wolf (*Canis lupus*) and jaguar (*Panthera onca*), and the Philippine pangolin (*Manis culionensis*) and Sunda pangolin (*M. javanica*). One of the betacoronaviruses with the highest sequence similarity to SARS-CoV-2 was isolated from Sunda pangolins [70]. Interestingly, pangolin burrows are known to be occupied by other animal species, including numerous bats [71].

Commonly hunted species in the top 10% of predictions include duiker (*Cephalophus zebra*, West Africa), warty pig (*Sus celebes*, Southeast Asia) and two deer (*Odocoileus*

*hemionus* and *O. virginianus*, Americas). The white-tailed deer (*O. virginianus*) was recently confirmed to transmit SARS-CoV-2 to conspecifics via aerosolized virus particles [72].

## (c) Bats

Our model identified 35 bat species within the 90th percentile of zoonotic capacity. Within the genus *Rhinolophus*, our model identified the large rufous horseshoe bat (*Rhinolophus*

*rufus*) as having the highest probability of zoonotic capacity (0.89). Rhinolophus rufus is a known natural host for bat betacoronaviruses [73] and a congener to three other horseshoe bats harbouring betacoronaviruses with high nucleotide sequence similarity to SARS-CoV-2 (approx. 92–96%) [6,74,75]. For these other three species, our model assigned a range of probabilities for SARS-CoV-2 zoonotic capacity (*Rhinolophus affinis* (0.58), *R. malayanus* (0.70) and *R. shameli* (0.71)) and also predicted relatively high probabilities for two congeners, *Rhinolophus acuminatus* (0.84) and *R. macrotis* (0.70). These predictions agree with recent experiments demonstrating efficient viral binding of SARS-CoV-2 RBD for *R. macrotis* [76] and confirmation of SARS-CoV-2-neutralizing antibodies in field-caught *R. acuminatus* harbouring a closely related betacoronavirus [77].

Our model also identified 17 species in the genus *Pteropus* (flying foxes) with high probabilities of zoonotic capacity for SARS-CoV-2. Some of these species are confirmed reservoirs of other zoonotic viruses (e.g. henipaviruses in *P. lylei*, *P. vampyrus*, *P. conspicillatus* and *P. alecto*), with Southeast Asia also having the most mammal species with a high predicted zoonotic capacity (figure 4). Annual outbreaks attributed to spillover transmission from bats illustrate a persistent epizootic risk to humans [78–80] and confirm that gaps in systematic surveillance of zoonotic viruses, including betacoronaviruses, remain an urgent priority (e.g. [81]).

## (d) Rodents

Our model identified 76 rodent species with high zoonotic capacity. Among these are the deer mouse (*Peromyscus maniculatus*) and white-footed mouse (*P. leucopus*), which are reservoirs for multiple zoonotic pathogens and parasites in North America [82–84]. Experimental infection, viral shedding and sustained intraspecific transmission of SARS-CoV-2 were recently confirmed for *P. maniculatus* [65,66]. Also in the top 10% were two rodents considered to be human commensals whose geographic ranges are expanding due to human activities: *Rattus argentiventer* (0.84) and *R. tiomanicus* (0.79) (electronic supplementary material, file S1) [85–87]. It is notable that many of these rodent species are preyed upon by carnivores, such as the red fox (*Vulpes vulpes*) or domestic cats (*Felis catus*) who themselves were predicted to have high zoonotic capacity by our model.

## (e) Strengthening predictive capacity for zoonoses

While there was a wide agreement between our model predictions and empirical studies, examining mismatches between experimental results and model-generated predictions focuses attention on characterizing what factors underlie these disconnects. For instance, this study and others predicted that pigs (*Sus scrofa*) would be susceptible to SARS-CoV-2 (figure 1), but these predictions have not been supported by whole-animal inoculations [38,69]. Similarly, SARS-CoV-2 replicated in adult cattle but onward transmission has not been observed *in vivo* [61].

Disconnects between real-world observations, *in vivo* experimental results, and *in silico* predictions of zoonotic capacity offer insight upon which to iterate further study. For instance, mismatches may arise because host susceptibility and transmission capacity are necessary but not sufficient for zoonotic risk to be realized in natural settings. These processes are embedded in a broader ecological context that impacts host susceptibility, intra-host infection dynamics (latency, recrudescence, tolerance) and viral persistence that collectively determine where and when viral shedding and spillover will occur [88–91]. Infection processes also depend strongly on the cellular environments in which cell entry and viral replication take place (e.g. the presence of key proteases [7]), and on host immunogenicity [91], factors which are themselves influenced by the environment [92]. Insofar as data limitations preclude perfect computational predictions of zoonotic capacity (e.g. limited ACE2 sequences, crystal structures or trait data), laboratory experiments are also limited. For SARS-CoV-2 and other host–pathogen systems, animals that are readily infected in the laboratory may be less susceptible elsewhere (ferrets in the laboratory versus mixed results in ferrets as pets [69,93,94]; rabbits in the laboratory versus rabbits as pets [95,96]). Moreover, wildlife hosts confirmed to shed multiple zoonotic viruses in nature (e.g. bats [97]) can be much less tractable for whole-animal laboratory investigations (for instance, requiring high biosecurity containment and very limited sample sizes). While laboratory experiments are critical for understanding mechanisms of pathogenesis and disease, without field surveillance they offer imperfect reflections of zoonotic capacity realized in natural settings.

These examples illustrate that there is no single methodology sufficient to understand and predict zoonotic transmission, for SARS-CoV-2 or any zoonotic pathogen. They also demonstrate the need to capitalize on underused or disconnected data sources, such as natural history collections, which are well-positioned to fill knowledge gaps about the spatial and temporal extents of animal hosts and their pathogens [98,99]. Integration of methods and data across biological scales creates avenues to more efficient iteration between computational predictions, laboratory experiments and targeted animal surveillance necessary to connect transmission mechanisms to the broader conditions underpinning zoonotic disease emergence in nature.

Data accessibility. All data, code and models required for recreating our structural modelling analysis with HADDOCK are available from Zenodo: https://doi.org/10.5281/zenodo.4517509 [52]. All data and code required for BRT modelling using species trait data are available from GitHub: https://github.com/HanLabDiseaseEcology/zoonotic_capacity. Additional methods, tables, figures and files of our model predictions can be found in the supplemental material of this article and on Figshare: https://doi.org/10.25390/caryinstitute.c.5293339.v5 [20].

Authors' contributions. I.R.F., A.A.C. and B.A.H. were responsible for conceptualization, data curation, formal analysis, investigation, methodology, project administration, software, and writing, reviewing and editing the manuscript. J.P.G.L.M.R. contributed additional analysis. J.P.G.L.M.R., A.V. and B.A.H. contributed funding. A.V. and A.A.C. created all figures. All authors gave final approval for publication and agreed to be held accountable for the work performed therein.

Competing interests. We declare we have no competing interests.

Funding. This work was supported by the NSF EEID program (grant no. DEB 1717282), DARPA PREEMPT program (grant no. D18AC00031), CREATE-NEO, a member of the NIH NIAID CREID program (grant no. 1U01 AI151807-01) and the NVIDIA Corporation GPU grant program (B.A.H.); by the NSF Polar program (grant no. OPP 1935870 and 1947040) (A.V.); and by NIH NIGMS (grant no. R35GM122543) (J.P.G.L.M.R.).

Acknowledgements. We are grateful for discussions with Drs. Alexandre Bonvin, Dennis Bente, Susan Hafenstein, Kathryn Hanley, Hyunwook Lee, Colin Parrish and John Paul Schmidt about various components of this project.

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
