## [Peer Review File · Proceedings of the Royal Society B: Biological Sciences]

Review History

RSPB-2021-1651.R0 (Original submission)

Review form: Reviewer 1

Recommendation

Accept with minor revision (please list in comments)

Scientific importance: Is the manuscript an original and important contribution to its field?

Excellent

General interest: Is the paper of sufficient general interest?

Good

Quality of the paper: Is the overall quality of the paper suitable?

Good

Is the length of the paper justified?

Yes

Should the paper be seen by a specialist statistical reviewer?

No

Do you have any concerns about statistical analyses in this paper? If so, please specify them explicitly in your report.

No

It is a condition of publication that authors make their supporting data, code and materials available - either as supplementary material or hosted in an external repository. Please rate, if applicable, the supporting data on the following criteria.

Is it accessible?

Yes

Is it clear?

Yes

Is it adequate?

Yes

Do you have any ethical concerns with this paper?

No

Comments to the Author

Overall, I thought this was a well written and very interesting paper, predicting the potential of different mammal species to transmit SARS-CoV-2 and potentially act as infection reservoirs in the future. I had only relatively minor comments:

Introduction - define SARS-CoV-2 on first use

Introduction - "ACE2 is ubiquitous across chordates, likely because of its role in highly conserved physiological pathways [38]." - think would be useful to briefly define the key physiological pathways here.

Introduction - "Our analyses are based on the initial dominant SARS-CoV-2 variant in humans, but these methods can be readily adjusted to enable host range predictions for new variants as their hACE2-RBD crystal structures become available." - Think these needs a little bit more explanation - first what is the initial dominant SARS-CoV2 in humans - specify? The initial wuhan strain lacked D614G in spike, but this rapidly became dominant. Secondly, most of the variants of concern now (which are the dominant ones in circulation - alpha, delta etc) have deletions at key positions in the spike protein affecting structure - is there really no preliminary predicted structures for these for which you could use (unlikely true crystal structures available) - this would inevitably lead to analysis of whether variants of concern are more likely to transmit/be sustained within non-human populations. I think the paper is strong as it is, but think this point needs to be discussed/addressed more.

Methods - Given that non-mammal hosts have yet to be confirmed as both susceptible and capable of onward transmission of SARS-CoV-2, we focused our modeling on mammals (for descriptions and results from additional uninformative models, including on all vertebrates, see Supplementary Methods and Supplementary Table 4). - the first line of results suggest Aves were also evaluated "The majority of sequences belonged to the classes Actinopterygii (22.1%), Aves (23.3%), and Mammalia (46.6%). We predicted binding strength for 299 vertebrates, including 142 mammals." Were Aves included in the other analyses - but not the zoonotic potential? Would it be possible to easily expand to include Aves in the future - perhaps a point for the discussion.

Discussion - "Our model predicted high zoonotic capacity in 20 out of 21 species in the primate genus *Macaca*, which comprise the majority of live-traded primates."

Which is the one species not predicted? any hypotheses as to why it is not?

Discussion - the discussion felt some what like the results - with multiple sections recapping/stating model prediction summaries - however, overall I found the paper to be well written and an easy read - if space needed to be trimmed I suspect it could be lost here

Github code - this lacks any sort of ReadMe as to what the code is, how to use/run it, and an example data set - think these are key requirements.

Reptiles - an noticed reptiles in the suppinfo - there was an early SARS-CoV-2 paper suggesting snakes as the reservoir for the virus - do any of your results suggest/refute this - the study was quite widely criticised e.g. see (with links to the original):

<https://virological.org/t/ncov-2019-codon-usage-and-reservoir-not-snakes-v2/339>

Review form: Reviewer 2

Recommendation

Accept with minor revision (please list in comments)

Scientific importance: Is the manuscript an original and important contribution to its field?

Good

General interest: Is the paper of sufficient general interest?

Excellent

Quality of the paper: Is the overall quality of the paper suitable?

Good

Is the length of the paper justified?

Yes

Should the paper be seen by a specialist statistical reviewer?

No

Do you have any concerns about statistical analyses in this paper? If so, please specify them explicitly in your report.

No

It is a condition of publication that authors make their supporting data, code and materials available - either as supplementary material or hosted in an external repository. Please rate, if applicable, the supporting data on the following criteria.

Is it accessible?

Yes

Is it clear?

Yes

Is it adequate?

Yes

Do you have any ethical concerns with this paper?

No

Comments to the Author

This is a well written and interesting manuscript. In this study the authors model interactions between SARS-Cov-2 and the cell receptors of wild animal hosts based on protein secondary structure inferred from amino acid sequences. They then use machine learning (BRT analyses) to predict the strength of these interactions based on trait variation among species, and to infer the zoonotic risk that various host species present. I think it's a clever study that will be of wide interest to readers. I would also love to see more collaborations like this between molecular biologists and disease ecologists. However, I do have a few concerns on reading the methods and results closely.

1. It is unclear how well traits predict expected binding strength.

The BRT analysis shows that the models of binding affinity may be mostly fitting noise (supplementary Table 4). In Table 4, and for mammals, there are two models that relate trait variation to binding affinity. In one the response variable is the charge of Amino Acid residue 30. This model is quite good, with a corrected AUC of over 0.8. However, in their supplemental methods the authors indicate (page 5):

"Our trait-based models predicted nearly all mammal species to have a negative charge at amino acid 30, thus the results of this model were relatively uninformative except to identify a few species with particularly low risk of binding SARS-CoV-2."

So that leaves the other model, the one of "binding strength" as the more important one. It has a corrected AUC score of only about 10%, and in fact the AUC is 50% higher on the training data than the test data. This often indicates overfit/ a model fit largely to noise.

This does not strike me as a critical flaw, since presumably predicted binding affinity was simply an additional predictor used in the model of zoonotic capacity (i.e., in addition to morphological and ecological traits). I would assume that a model based on traits alone is still pretty good, and so this is a valuable study regardless. However, I think this is an issue the authors need to discuss. It might also be worth testing how much the fit of a model that includes binding affinity is improved compared to a model fit to traits alone.

2. Methods and results are not as clear as they should be.

The methods and results are somewhat unclear in general. Even after reading the supplemental methods closely, I am not 100% sure what variables were included in the final models shown in Table 4. I see two lists of traits (Table 2 and Table 3), and I'm not sure whether these are all traits considered, or all traits that made it past the selection criteria into the final models shown in Table 4. A table showing the relative influence scores of all variables in the final models would greatly clarify the results.

3. Incorporating phylogeny might improve model fits.

Including phylogeny as a predictor might help improve binding strength model. For example the authors find that two artiodactyls (members of the same broad clade) that their model predicts should present a high risk of transmission for SARS (pigs and cows) do not appear to be competent to transmit the virus in experimental studies.

Bianchini and Morrisey (2020) provide an illustration of how phylogenetic eigenvectors can be added into a BRT analysis. Of course they also concluded, counterintuitively, that phylogeny wasn't very important because it only explained 26% of deviance in the response variable above

and beyond that explained by species traits . . .

Bianchini, K., & Morrissey, C. A. (2020). Species traits predict the aryl hydrocarbon receptor 1 (AHR1) subtypes responsible for dioxin sensitivity in birds. *Scientific reports*, 10(1), 1-11. Even if the authors don't wish to included phylogeny directly, it seems that taxonomy might have some important information. The authors do mention collecting data from taxonomic databases, so perhaps taxonomy was included (if that is the case ignore this point, but see point 2).

At the very least, I think some readers may wonder why phylogeny was not considered, since many methods for imputing missing species trait values rely heavily on it and there are good trees for mammals available.

Decision letter (RSPB-2021-1651.R0)

28-Sep-2021

Dear Dr Han:

Your manuscript has now been peer reviewed and the reviews have been assessed by an Associate Editor. The reviewers' comments (not including confidential comments to the Editor) and the comments from the Associate Editor are included at the end of this email for your reference. As you will see, the reviewers have raised some concerns with your manuscript and we would like to invite you to revise your manuscript to address them.

Research ethics:

Use of animals and field studies:

It is a condition of publication that you make available the data and research materials supporting the results in the article. Please see our Data Sharing Policies (<https://royalsociety.org/journals/authors/author-guidelines/#data>). Datasets should be deposited in an appropriate publicly available repository and details of the associated accession number, link or DOI to the datasets must be included in the Data Accessibility section of the article (<https://royalsociety.org/journals/ethics-policies/data-sharing-mining/>). Reference(s) to datasets should also be included in the reference list of the article with DOIs (where available).

Please submit a copy of your revised paper within three weeks. If we do not hear from you within this time your manuscript will be rejected. If you are unable to meet this deadline please let us know as soon as possible, as we may be able to grant a short extension.

Best wishes,

Professor Hans Heesterbeek

Associate Editor

Board Member: 1

Comments to Author:

This is a very interesting study that uses a combination of molecular structural modeling and ecological data to make predictions about the potential for various mammal species to be effective transmitters of SARS-CoV-2. It is exactly this sort of forwarding looking analysis that holds a lot of promise for identifying high risk zoonotic scenarios. Both referees found the study very interesting as well and both provided a number of helpful comments to further strengthen the manuscript. I think this will make a very nice contribution to Proc B if the authors can address the comments raised by the referees.

Reviewer(s)' Comments to Author:

Referee: 1

Comments to the Author(s)

Overall, I though this was a well written and very interesting paper, predicting the potential of different mammal species to transmit SARS-CoV-2 and potentially act as infection reservoirs in the future. I had only relatively minor comments:

Introduction - define SARS-CoV-2 on first use

Introduction - "ACE2 is ubiquitous across chordates, likely because of its role in highly conserved physiological pathways [38]." - think would be useful to briefly define the key physiological pathways here.

Introduction - "Our analyses are based on the initial dominant SARS-CoV-2 variant in humans, but these methods can be readily adjusted to enable host range predictions for new variants as their hACE2-RBD crystal structures become available." - Think these needs a little bit more explanation - first what is the initial dominant SARS-CoV2 in humans - specify? The initial wuhan strain lacked D614G in spike, but this rapidly became dominant. Secondly, most of the variants of concern now (which are the dominant ones in circulation - alpha, delta etc) have deletions at key positions in the spike protein affecting structure - is there really no preliminary predicted structures for these for which you could use (unlikely true crystal structures available) - this would inevitably lead to analysis of whether variants of concern are more likely to transmit/be sustained within non-human populations. I think the paper is strong as it is, but think this point needs to be discussed/addressed more.

Methods - Given that non-mammal hosts have yet to be confirmed as both susceptible and capable of onward transmission of SARS-CoV-2, we focused our modeling on mammals (for descriptions and results from additional uninformative models, including on all vertebrates, see Supplementary Methods and Supplementary Table 4). - the first line of results suggest Aves were also evaluated "The majority of sequences belonged to the classes Actinopterygii (22.1%), Aves (23.3%), and Mammalia (46.6%). We predicted binding strength for 299 vertebrates, including 142 mammals." Were Aves included in the other analyses - but not the zoonotic potential? Would it be possible to easily expand to include Aves in the future - perhaps a point for the discussion.

Discussion - "Our model predicted high zoonotic capacity in 20 out of 21 species in the primate genus *Macaca*, which comprise the majority of live-traded primates." Which is the one species not predicted? any hypotheses as to why it is not?

Discussion - the discussion felt some what like the results - with multiple sections recapping/stating model predcition summaries - however, overall I found the paper to be well written and an easy read - if space needed to be trimmed I suspect it could be lost here

Github code - this lacks any sort of ReadMe as to what the code is, how to use/run it, and an example data set - think these are key requirements.

Reptiles - an noticed reptiles in the supinfo - there was an early SARS-CoV-2 paper suggesting snakes as the reservoir for the virus - do any of your results suggest/refute this - the study was quite widely criticised e.g. see (with links to the original):

<https://virological.org/t/ncov-2019-codon-usage-and-reservoir-not-snakes-v2/339>

Referee: 2

Comments to the Author(s)

This is a well written and interesting manuscript. In this study the authors model interactions between SARS-Cov-2 and the cell receptors of wild animal hosts based on protein secondary structure inferred from amino acid sequences. They then use machine learning (BRT analyses) to predict the strength of these interactions based on trait variation among species, and to infer the zoonotic risk that various host species present. I think it's a clever study that will be of wide interest to readers. I would also love to see more collaborations like this between molecular biologists and disease ecologists. However, I do have a few concerns on reading the methods and results closely.

1. It is unclear how well traits predict expected binding strength.

The BRT analysis shows that the models of binding affinity may be mostly fitting noise (supplementary Table 4). In Table 4, and for mammals, there are two models that relate trait variation to binding affinity. In one the response variable is the charge of Amino Acid residue 30. This model is quite good, with a corrected AUC of over 0.8. However, in their supplemental methods the authors indicate (page 5):

"Our trait-based models predicted nearly all mammal species to have a negative charge at amino acid 30, thus the results of this model were relatively uninformative except to identify a few species with particularly low risk of binding SARS-CoV-2."

So that leaves the other model, the one of "binding strength" as the more important one. It has a corrected AUC score of only about 10%, and in fact the AUC is 50% higher on the training data than the test data. This often indicates overfit/ a model fit largely to noise.

This does not strike me as a critical flaw, since presumably predicted binding affinity was simply an additional predictor used in the model of zoonotic capacity (i.e., in addition to morphological and ecological traits). I would assume that a model based on traits alone is still pretty good, and so this is a valuable study regardless. However, I think this is an issue the authors need to discuss. It might also be worth testing how much the fit of a model that includes binding affinity is improved compared to a model fit to traits alone.

2. Methods and results are not as clear as they should be.

The methods and results are somewhat unclear in general. Even after reading the supplemental methods closely, I am not 100% sure what variables were included in the final models shown in Table 4. I see two lists of traits (Table 2 and Table 3), and I'm not sure whether these are all traits considered, or all traits that made it past the selection criteria into the final models shown in Table 4. A table showing the relative influence scores of all variables in the final models would greatly clarify the results.

3. Incorporating phylogeny might improve model fits.

Including phylogeny as a predictor might help improve binding strength model. For example the authors find that two artiodactyls (members of the same broad clade) that their model predicts should present a high risk of transmission for SARS (pigs and cows) do not appear to be competent to transmit the virus in experimental studies.

Bianchini and Morrissey (2020) provide an illustration of how phylogenetic eigenvectors can be added into a BRT analysis. Of course they also concluded, counterintuitively, that phylogeny wasn't very important because it only explained 26% of deviance in the response variable above and beyond that explained by species traits . . .

Bianchini, K., & Morrissey, C. A. (2020). Species traits predict the aryl hydrocarbon receptor 1 (AHR1) subtypes responsible for dioxin sensitivity in birds. *Scientific reports*, 10(1), 1-11.

Even if the authors don't wish to included phylogeny directly, it seems that taxonomy might have some important information. The authors do mention collecting data from taxonomic databases, so perhaps taxonomy was included (if that is the case ignore this point, but see point 2).

At the very least, I think some readers may wonder why phylogeny was not considered, since many methods for imputing missing species trait values rely heavily on it and there are good trees for mammals available.

Author's Response to Decision Letter for (RSPB-2021-1651.R0)

See Appendix A.

Decision letter (RSPB-2021-1651.R1)

25-Oct-2021

Dear Dr Han

I am pleased to inform you that your manuscript entitled "Predicting the zoonotic capacity of mammals to transmit SARS-CoV-2" has been accepted for publication in *Proceedings B*.

Data Accessibility section

Open Access

You are invited to opt for Open Access, making your freely available to all as soon as it is ready for publication under a CC BY licence. Our article processing charge for Open Access is £1700. Corresponding authors from member institutions

Paper charges

Sincerely,

Professor Hans Heesterbeek

Appendix A

19 October 2021

Dear Dr. Heesterbeek,

Many thanks for the opportunity to revise and resubmit our manuscript, "*Predicting the zoonotic capacity of mammals to transmit SARS-CoV-2.*" The comments from peer reviewers were very helpful in directing our clarifications of the modeling methods and related text throughout the main manuscript and our supplementary materials. A tracked changes version of the main paper is appended to the bottom of this document as requested. In our replies below, we reference the line numbers corresponding to the tracked changes version of the document.

We hope that you will now find our paper suitable for publication in Proceedings B.

Sincerely,

Barbara Han

Associate Editor
Board Member: 1

This is a very interesting study that uses a combination of molecular structural modeling and ecological data to make predictions about the potential for various mammal species to be effective transmitters of SARS-CoV-2. It is exactly this sort of forwarding looking analysis that holds a lot of promise for identifying high risk zoonotic scenarios. Both referees found the study very interesting as well and both provided a number of helpful comments to further strengthen the manuscript. I think this will make a very nice contribution to Proc B if the authors can address the comments raised by the referees.

Thank you!

Referee: 1

Overall, I thought this was a well written and very interesting paper, predicting the potential of different mammal species to transmit SARS-CoV-2 and potentially act as infection reservoirs in the future. I had only relatively minor comments:

Introduction - define SARS-CoV-2 on first use

Now defined in the first sentence of the revised Introduction.

Introduction - "ACE2 is ubiquitous across chordates, likely because of its role in highly conserved physiological pathways [38]." - think would be useful to briefly define the key physiological pathways here.

Thanks for this suggestion. We have added a couple of examples, including citations to work describing the role of ACE2 in the regulation of blood pressure, salt, and water (lines 115-116 of the revision)

Introduction - "Our analyses are based on the initial dominant SARS-CoV-2 variant in humans, but these methods can be readily adjusted to enable host range predictions for new variants as their hACE2-RBD crystal structures become available." - Think these needs a little bit more explanation - first what is the initial dominant SARS-CoV2 in humans - specify? The initial wuhan strain lacked D614G in spike, but this rapidly became dominant.

For all analyses, we used the crystal structure of SARS-CoV-2 RBD bound to a human ACE2 receptor (PDB ID: 6m0j) (the "initial dominant strain"). We previously included this information in the supplementary, but we have now moved the PDB ID and reference to the main text (reference number 48, and line 156 of the revision). As new crystal structures become available, the HADDOCK structural modeling + trait modeling pipeline can be rerun to generate new species predictions about zoonotic capacity based on updated HADDOCK scores.

Secondly, most of the variants of concern now (which are the dominant ones in circulation - alpha, delta etc) have deletions at key positions in the spike protein affecting structure - is there really no preliminary predicted structures for these for which you could use (unlikely true crystal structures available) - this would inevitably lead to analysis of whether variants of concern are more likely to transmit/be sustained within non-human populations. I think the paper is strong as it is, but think this point needs to be discussed/addressed more.

We are unaware of any crystal structures available for these newer variants. Some of the observed deletions appear in loops that are distantly located from the receptor binding domain. Other variations include point mutations within the receptor binding domain, some of which could be consequential for species predictions. For instance, a single mutation in the Asp/Glu has the potential to cause large differences in HADDOCK scores. In contrast to other available structural modeling approaches, HADDOCK accounts for many of the structural differences arising from deletions and point mutations by flexibly allowing for contacts across the binding interface (ambiguous interaction constraints) (HADDOCK 2.4 Manual, <https://www.bonvinlab.org/software/haddock2.4/introduction/>). It is still possible that multiple simultaneous mutations could lead to substantial differences in structure (as exemplified by the variants of concern brought up in this comment), but the effects of these combined mutations remain quite difficult to predict (and therefore, difficult to account for in computational approaches). We have added a few sentences to the end of the first paragraph of the Methods section (section header: "*Structural Modeling of ACE2 orthologs bound to SARS-CoV-2 spike*") to communicate how our modeling pipeline could be applied to explore binding strengths of variants and generate predictions about host range.

Methods - Given that non-mammal hosts have yet to be confirmed as both susceptible and capable of onward transmission of SARS-CoV-2, we focused our modeling on mammals (for descriptions and results from additional uninformative models, including on all vertebrates, see Supplementary Methods and Supplementary Table 4). - the first line of results suggest Aves were also evaluated "The majority of sequences belonged to the classes Actinopterygii (22.1%), Aves (23.3%), and Mammalia (46.6%). We

predicted binding strength for 299 vertebrates, including 142 mammals." Were Aves included in the other analyses - but not the zoonotic potential? Would it be possible to easily expand to include Aves in the future - perhaps a point for the discussion.

Yes, that is correct - Aves were included in the vertebrate wide analysis, but excluded from the main analysis of the paper which focused on Mammalia. The reviewer raises an interesting possibility of expanding these analyses to consider birds and mammals together. Such an expansion would first require looking for other available trait data specific to birds to create a more comprehensive dataset for this group. Then we would train a machine learning model to predict a coarse grained label of binding (strong vs. weak) using these avian and mammalian traits. A probable issue here is a lack of ACE2 sequences for birds. Of the bird species with ACE2 sequences available, only 9 appear above the threshold we used to model zoonotic capacity, which could lead to highly unbalanced labels for a small dataset.

Discussion - "Our model predicted high zoonotic capacity in 20 out of 21 species in the primate genus *Macaca*, which comprise the majority of live-traded primates." Which is the one species not predicted? any hypotheses as to why it is not?

We clarified the wording here to read "The *Macaca* genus comprised the majority of live-traded primates. Our model predicted high zoonotic capacity for nearly all *Macaca* species (20/21 species, with all species within the top 10% of predictions except *Macaca assemensis*)".

The predicted probabilities of *Macaca* ranged from 0.988 (*Macaca fascicularis*) to 0.788 (*M. assamensis*). While all of the *Macaca* were predicted above 0.5 probability in all 50 bootstraps performed, the only *Macaca* that was not within the top 10 percent was *M. assamensis*. It could be that the lower score for this species reflects a larger distance in trait space from positively labeled primates (e.g., *M. fascicularis*).

Discussion - the discussion felt some what like the results - with multiple sections recapping/stating model prediction summaries - however, overall I found the paper to be well written and an easy read - if space needed to be trimmed I suspect it could be lost here

Thanks very much. Our intention was to logically group results together to make them more accessible to researchers who are best positioned to conduct follow up testing, and to point out why certain predictions are noteworthy. An unintended consequence was that our discussion ended up feeling like a "laundry list". We hope this tradeoff between accessibility and prose is still worthwhile.

Github code - this lacks any sort of ReadMe as to what the code is, how to use/run it, and an example data set - think these are key requirements.

Thanks for this feedback. We elaborated on the existing Readme file in the github repo (https://github.com/HanLabDiseaseEcology/zoonotic_capacity) to provide more information about the code, how to use it, and to identify the datasets in the repo that can be used to 1) reproduce the zoonotic capacity model, and 2) to make predictions for 5,400 mammals with the fitted model. In the `zoonotic_capacity.Rmd` file, we have also added numbering to the code chunks, additional commenting throughout, and we 'commented

out' some sections of the code that were specific to the less informative models to make our overall process a bit easier to follow.

Reptiles - an noticed reptiles in the supinfo - there was an early SARS-CoV-2 paper suggesting snakes as the reservoir for the virus - do any of your results suggest/refute this - the study was quite widely criticised e.g. see (with links to the original): <https://virological.org/t/ncov-2019-codon-usage-and-reservoir-not-snakes-v2/339>

The results from our structural modeling using HADDOCK do not support the suggestion that snakes (and other reptiles) are reservoirs for the virus. Supplementary Figure 2B shows this fairly well in a box plot of HADDOCK scores grouped by class. All of the predicted binding strengths of reptiles with ACE2 information are below the threshold of susceptible species that can transmit to conspecifics and are typically lower than most other species.

Referee: 2

This is a well written and interesting manuscript. In this study the authors model interactions between SARS-Cov-2 and the cell receptors of wild animal hosts based on protein secondary structure inferred from amino acid sequences. They then use machine learning (BRT analyses) to predict the strength of these interactions based on trait variation among species, and to infer the zoonotic risk that various host species present. I think it's a clever study that will be of wide interest to readers. I would also love to see more collaborations like this between molecular biologists and disease ecologists. However, I do have a few concerns on reading the methods and results closely.

1. It is unclear how well traits predict expected binding strength.

The BRT analysis shows that the models of binding affinity may be mostly fitting noise (supplementary Table 4). In Table 4, and for mammals, there are two models that relate trait variation to binding affinity. In one the response variable is the charge of Amino Acid residue 30. This model is quite good, with a corrected AUC of over 0.8. However, in their supplemental methods the authors indicate (page 5):

"Our trait-based models predicted nearly all mammal species to have a negative charge at amino acid 30, thus the results of this model were relatively uninformative except to identify a few species with particularly low risk of binding SARS-CoV-2."

So that leaves the other model, the one of "binding strength" as the more important one. It has a corrected AUC score of only about 10%, and in fact the AUC is 50% higher on the training data than the test data. This often indicates overfit/ a model fit largely to noise.

The large discrepancy in evaluation statistics is because the model of binding strength predicts a continuous variable (not binary), so the evaluation statistic is a pseudo-R² value rather than an AUC. This wasn't obvious given how the table is constructed, so we added an additional column to specify when evaluation statistics are AUC vs. pseudo-R² in Supplementary Table 4.

This does not strike me as a critical flaw, since presumably predicted binding affinity was simply an additional predictor used in the model of zoonotic capacity (i.e., in addition to morphological and ecological traits). I would assume that a model based on traits alone is still pretty good, and so this is a valuable study regardless. However, I think this is an issue the authors need to discuss. It might also be worth testing how much the fit of a model that includes binding affinity is improved compared to a model fit to traits alone.

The model using traits alone (mammal zoonotic capacity model) does indeed perform well (AUC 0.725) without exhibiting uninformative classification issues we observed for the AA30 model. The reviewer is correct that this model did not include estimated binding strength as a predictor. This is because we used traits to directly predict binarized binding strength scores across species (including binding strength as a predictor variable would be circular). Binary scores of zoonotic capacity were obtained by comparing estimated binding strengths across species against HADDOCK score = -129. This threshold value was determined empirically, falling in-between the estimated binding strengths for domesticated cats and domesticated pigs: *Felis catus* had the weakest estimated binding for which conspecific transmission has been observed *in vivo*, and *Sus scrofa* had the strongest estimated binding and no evidence of conspecific transmission observed *in vivo*.

To clarify this further, we added a short definition of zoonotic capacity, in relation to estimated binding strength, to the caption for Supplementary Table 4.

2. Methods and results are not as clear as they should be.

The methods and results are somewhat unclear in general. Even after reading the supplemental methods closely, I am not 100% sure what variables were included in the final models shown in Table 4. I see two lists of traits (Table 2 and Table 3), and I'm not sure whether these are all traits considered, or all traits that made it past the selection criteria into the final models shown in Table 4. A table showing the relative influence scores of all variables in the final models would greatly clarify the results.

Thanks very much for these suggestions. We reviewed the Results section and our Supplementary Methods and added a section to the beginning of the Supp Methods to clarify what models we ran, and which variables were ultimately included in the modeling for each model. We also constructed a new table that consolidates Supp Tables 2 and 3, which now contains all of the variables we considered for vertebrate and mammal models and includes % coverage for each variable. We also added to the modeling section of the Supplementary Methods clarifying that the variables included in the newly consolidated Supplementary Table 2 are the ones included in all modeling, and specifying that we removed variables with near zero variation.

Thanks also for pointing out that we excluded a plot of relative importance scores. We now include this as a new Supplementary Figure 7. The variables showing up in this final plot (for mammals) also represent those that were included in the final model (i.e., variables not in this plot were excluded from the model due to near zero variation). We defined fields to have near zero variation if the ratio of the most common value in the field to the second most common value was above 95/5.

3. Incorporating phylogeny might improve model fits.

Including phylogeny as a predictor might help improve binding strength model. For example the authors find that two artiodactyls (members of the same broad clade) that their model predicts should present a high risk of transmission for SARS (pigs and cows) do not appear to be competent to transmit the virus in experimental studies.

Bianchini and Morrissey (2020) provide an illustration of how phylogenetic eigenvectors can be added into a BRT analysis. Of course they also concluded, counterintuitively, that phylogeny wasn't very important because it only explained 26% of deviance in the response variable above and beyond that explained by species traits . . .

Bianchini, K., & Morrissey, C. A. (2020). Species traits predict the aryl hydrocarbon receptor 1 (AHR1) subtypes responsible for dioxin sensitivity in birds. *Scientific reports*, 10(1), 1-11.

Even if the authors don't wish to included phylogeny directly, it seems that taxonomy might have some important information. The authors do mention collecting data from taxonomic databases, so perhaps taxonomy was included (if that is the case ignore this point, but see point 2).

At the very least, I think some readers may wonder why phylogeny was not considered, since many methods for imputing missing species trait values rely heavily on it and there are good trees for mammals available.

We included phylogenetic information directly into all of our models. For the mammal-specific model, which formed the bulk of the main paper, we included phylogeny as binary variables of mammalian families. For the vertebrate-wide model (in supplementary) we included orders as binary variables.

Our previous experience with phylogenetic eigenvectors echoes the results of the Bianchini and Morrissey paper cited above: eigenvectors does not provide much additional predictive power compared to the simpler approach of adding taxonomic variables as binary predictors, which also have the advantage of being easier to interpret. We therefore did not apply eigenvectors here, and defaulted to this simpler method of accounting for phylogeny as predictor variables given our previous success with this approach. We have revised line 173 of our Methods to better communicate that we included phylogeny in all analyses.

In general, comments from reviewer 2 helped us to make clarifying edits throughout the revised manuscript, and to conceptual figure (Figure 2) and the legend. Many thanks for the careful reading and all the valuable comments.